# FedAgent-HPO: Agentic Hyperparameter Optimization for Personalized Federated Learning

## Abstract

Hyperparameter optimization (HPO) in federated learning (FL) is especially challenging because heterogeneous clients and non-IID data make global tuning unreliable and expensive. Existing wrapper-based methods (e.g., grid search, random search, etc.) often assess numerous configurations through retraining, which is prohibitively costly in communication and computation. We propose **FedAgent-HPO**, a personalized and explicable framework that reframes HPO in FL as an agentic reasoning task. The framework introduces two cooperating agents. The Hyperparameter Agent generates personalized client and server configurations using local history and peer-aware signals. The Analyzer Agent dynamically adapts the search space, improving efficiency. FedAgent-HPO operates online during training and uses an asynchronous CPU–GPU pipeline to overlap agent inference with model updates, reducing wall-clock time. Built on SplitFed, it adapts to diverse client resources while producing interpretable hyperparameter trajectories. Experiments on vision, language, and cross-silo benchmarks show that FedAgent-HPO improves accuracy by up to 8.8% under non-IID data and reduces training time through asynchronous execution.

## 1 Introduction

Federated Learning (FL) allows for collaborative training without the need to centralize data McMahan et al. (2017), yet its effectiveness depends largely on the sensitivity of hyperparameter (HP) choices Kuo et al. (2023). In heterogeneous networks, depending on a single global hyperparameter (HP) set can be unreliable. Clients differ not only in their data distributions but also in their computing capabilities, memory, and bandwidth. As a result, the optimal learning rate, batch size, optimizer, and proximal strength can vary significantly between clients. This has motivated personalized tuning—e.g., HPN Cheng et al. (2023)—as a key open direction in FL Kairouz et al. (2021). Even though FL has become a cornerstone of modern distributed machine learning, inherent statistical and systemic heterogeneity Li et al. (2020a) makes per-client HP policies particularly important.

Recent works adopt a tuning-while-training paradigm Jiang (2021); Luketina et al. (2022) that alternates between hyperparameter updates and federated learning. However, many techniques still rely on outer-loop wrappers (e.g., random/bayesian search or successive halving (SHA)) that evaluate entire FL runs. For example, FedEx Khodak et al. (2021) maintains a distribution over HPs using reward-based updates but still expends multiple full training cycles to explore. Similarly, FedPop Chen et al. (2025a) evolves populations of configurations across rounds. These wrappers incur high costs in computation and communication because they generally require many complete runs to achieve satisfactory accuracy. They also delay responsiveness and overlook practical device limitations such as memory, bandwidth, and throughput.

Our goal is to understand per-client hyperparameter (HP) policies with training that would (i) take into consideration statistical and resource variation; (ii) be in alignment with feasibility constraints, e.g., memory-safe batch sizes under a given model split; and (iii) improve overall accuracy and stability across heterogeneous clients. In addition to exploring the role of standard Federated Learning (FL), we will also investigate the effectiveness of Split Federated Learning (SplitFed) in addressing resource constraints by partitioning models between the client and server.

**Why agentic policies?** Grid/random search require multiple full FL runs—prohibitively costly when each run involves hundreds of clients and many communication rounds Bergstra & Bengio (2012); Khodak et al. (2021). Agent-based HPO instead adapts HPs online within a single run Liu et al. (2025), enabling per-client personalization, responsiveness to non-IID drift, and enforcement of resource constraints without the overhead of repeated training cycles.

We introduce **FedAgent-HPO**, a constrained dual-agent controller for personalized FL-HPO. An *Analyzer Agent* refines each client's feasible HP set using recent outcomes and peer summaries; an *HP Agent* proposes actions that are then projected onto the Analyzer's feasible set, guaranteeing constraint adherence. The controller conditions on each client's data and resource profile, making it naturally compatible with SplitFed partitioning—yet it also operates without SplitFed when resources permit. For stability with high personalization, we adopt the standard FedProx formulation Li et al. (2020b), in which clients approximately minimize a proximal-regularized local objective and the server aggregates updates via data-weighted averaging. On the systems side, we introduce a non-blocking, asynchronous agent–trainer architecture that overlaps LLM reasoning with training, effectively reducing overall training latency. Throughout, agents consume only compact scalar summaries (loss/accuracy, simple timing, and HPs used), not activations or raw data. Our main contributions are:

- We propose FedAgent-HPO, the first agent-based hyperparameter optimization framework for federated learning that utilizes LLM reasoning ability to carry out per-client dynamic hyperparameter tuning.

- An asynchronous CPU-GPU pipeline was built to decouple LLM-based reasoning from FL training, reducing computational bottlenecks and achieving up to 39% reduction in per-client training time.

- We combine the personalized HPO with the SplitFed architecture so that resource-constrained clients can be served without the loss of optimization effectiveness either in homogeneous or in heterogeneous environments.

- We demonstrate consistent improvement in accuracy across vision and natural language processing benchmarks, achieving up to 8.8% over state-of-the-art federated hyperparameter optimization baselines and highly non-IID distributions.

## 2 BACKGROUND AND RELATED WORKS

### 2.1 FEDERATED HYPERPARAMETER OPTIMIZATION

Hyperparameter optimization (HPO) in FL has been explored via both wrapper-based and online tuning paradigms. Early methods applied black-box searches such as random search Bergstra & Bengio (2012) and Hyperband Li & Jamieson (2018) directly to FL, treating each configuration as a full training run and incurring high computation and communication costs Yoon et al. (2021). To reduce overhead, integrated approaches emerged: FedEx adapts client learning rates via distributional updates Khodak et al. (2021), while FedPop uses an evolutionary population of configurations Chen et al. (2025a). Auto-FedRL leverages reinforcement learning for client-specific tuning Guo et al. (2022), and system-level methods jointly optimize hyperparameters with resource allocation Liu et al. (2022). Domain-driven strategies include heuristic adaptive tuning for energy prediction Toderean et al. (2025) and clustering-based electrical load forecasting Gholizadeh & Musilek (2021), which adapt settings to client groups based on task similarity. Despite these advances, most methods either rely on expensive wrapper loops, restrict search to global values, or lack principled personalization for heterogeneous clients Wang et al. (2023). Moreover, existing techniques largely act as black-box optimizers without interpretable reasoning, limiting trust and compliance in practical FL deployments Zhang & Yu (2024).

### 2.2 LLM-BASED AGENT OPTIMIZATION IN HPO

LLMs have opened a new direction for agent-based reasoning in optimization. AgentHPO Liu et al. (2025) demonstrated that LLM agents can iteratively propose, evaluate, and refine hyperparameters while providing natural-language explanations for their choices. Similarly, LLM Agent for

Hyper-Parameter Optimization Wang et al. (2025) applied an LLM-based reasoning workflow to sequential tuning tasks, and Optima Chen et al. (2025b) explored collaborative multi-agent structures for efficient decision-making in AutoML pipelines. Another study has also shown that stepwise reinforcement learning can enable LLM agents to acquire expert-level optimization behaviors Deng et al. (2024). However, these studies are conducted in conventional (centralized) ML settings and do not address the unique challenges of FL, where client heterogeneity, privacy constraints, and asynchronous updates impose stricter requirements.

Our proposed FedAgent-HPO bridges these gaps by uniting personalized federated hyperparameter optimization with LLM-based agent reasoning. In contrast to existing approaches, it supports asynchronous adaptation to avoid training stalls and integrates SplitFed to reduce client computation and enhance privacy. This yields an adaptive, efficient, and explainable federated HPO framework that overcomes the cost and transparency limitations of prior methods.

## 3 METHODOLOGY

We introduce FedAgent-HPO, a personalized hyperparameter optimization framework for federated learning in heterogeneous, resource-constrained environments. As illustrated in Figure 1, the system integrates LLM-based agents with a SplitFed architecture to adapt hyperparameters at per-client granularity. The workflow combines resource profiling with clustering, SplitFed training, asynchronous agent–trainer interaction, and FedProx-regularized aggregation; each component is detailed below.

### 3.1 PHASE 1: SPLITFED WITH RESOURCE-AWARE CLUSTERING

In conventional federated learning, each client trains the entire model locally. This assumption is brittle in practice because client devices differ widely in compute, memory, and network capacity; the same global configuration can be unstable for slow or memory-constrained clients and underutilize faster ones. These system constraints interact directly with hyperparameters (HPs): batch size, learning rate, momentum, and regularization exhibit different stability and convergence regimes depending on how much work a client can execute per step. To decouple these concerns, we adopt a SplitFed architecture in which the global network $f(\cdot; \theta)$ is partitioned as $f(\cdot; \theta) = f_s(f_c(\cdot; \theta_c); \theta_s)$. During training, client $i$ computes an intermediate representation

$$z_i = f_c(x_i; \theta_c),$$

sends $z_i$ to the server, the server completes the forward pass $\hat{y}_i = f_s(z_i; \theta_s)$, computes the loss $L(\hat{y}_i, y_i)$, and returns the gradient $\partial \ell / \partial z_i$ for the cut. The client then performs backpropagation through $f_c$ using the chain rule, i.e., it updates $\theta_c$ based on $(\partial \ell / \partial z_i)(\partial z_i / \partial \theta_c)$. This split allows low-resource clients to offload deeper layers while still participating stably in training, and it preserves a consistent interface across heterogeneous devices. *(Pseudocode for this phase provided in Appendix A.2 (Algorithm 1))*

*Resource-aware clustering.* To assign an appropriate split point once and keep it fixed for the rest of training, we profile each client at enrollment based on readily available signals—hardware capability (RAM and GPU availability) and a lightweight proxy for throughput (simple inference speed). Clients are grouped into clusters $\{C_1, \ldots, C_K\}$, and each cluster is assigned a cut depth that is feasible for its typical device profile. High-resource clusters compute more layers locally, enabling more aggressive local HPs (e.g., larger batches or faster schedules), while low-resource clusters offload deeper layers, making accuracy more sensitive to server-side HPs such as optimizer momentum or scheduler parameters. This one-time clustering stabilizes each client's operating point, and Phase 2 then personalizes hyperparameters on top of this split without changing the split itself.

Since our experiments do not rely on real heterogeneous devices, we *simulate* lightweight device and link profiles for each client at enrollment. As shown in Algorithm 1 (Step 1), each client $i$ is assigned a feature vector

$$\phi_i = \{\text{available\_memory}_i, \text{GPU\_availability}_i, \text{uplink\_bw}_i, \text{downlink\_bw}_i, \text{warmupStepTime}_i\},$$

where RAM/GPU availability model compute capability, uplink/downlink bandwidth capture communication quality, and warmup step time acts as a latency/throughput proxy. These values are

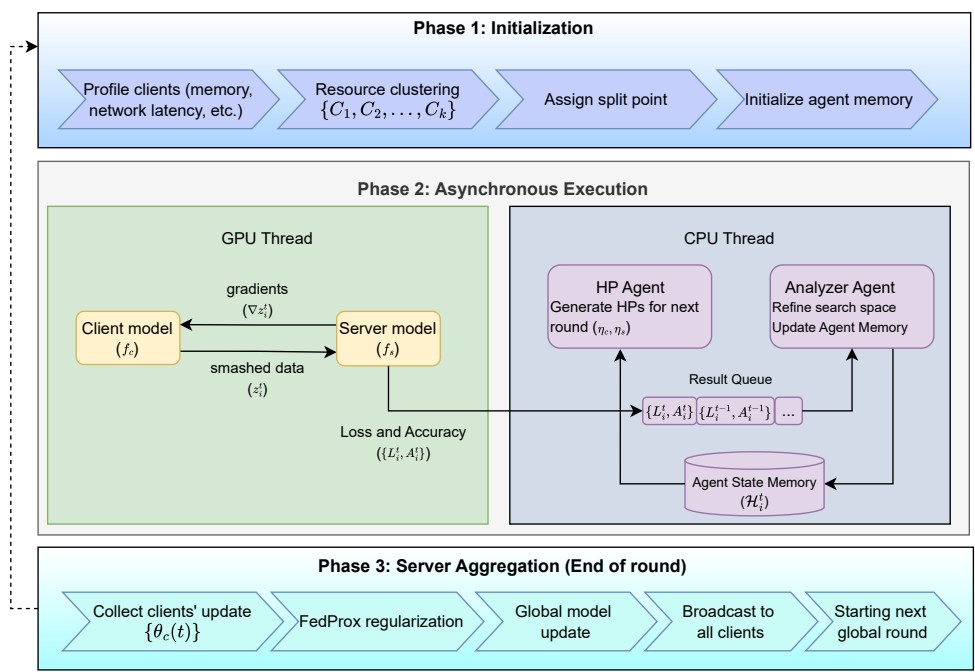

Figure 1: Overall architecture of FedAgent-HPO showing the dual-agent controller and its integration with SplitFed.

sampled from predefined ranges representing low-, mid-, and high-resource clients. We cluster the vectors $\{\phi_i\}$ into $K = 3$ resource tiers using $k$-means on normalized features. Each tier $C_k$ is assigned a split point $z_k \in Z$ that matches its typical bandwidth/latency characteristics, guided by activation size $|z_k|$ and per-batch traffic $2b|z_k|$ (Algorithm 1, Step 2). We set $R = 0$ in all experiments, so clustering is performed once at enrollment.

## 3.2 PHASE 2: ASYNCHRONOUS EXECUTION—DECOUPLING TRAINING AND REASONING

LLM-based HP reasoning introduces CPU/network latency that conflicts with GPU-bound split-FL training. Calling an LLM inside the training loop stalls GPUs, so we decouple training and reasoning using two concurrent threads and a non-blocking queue.

*GPU-bound training thread.* Each client performs forward/backward through its local submodel, exchanges cut-layer activations and gradients with the server, and completes its local round. Afterward, it pushes only compact scalar summaries (loss/accuracy, lightweight timing stats, and the HPs used) to the shared queue. No raw data, activations, or gradients are exposed, preserving the split-learning privacy boundary.

*CPU-bound agent thread.* Independently, the HP Agent and Analyzer consume queue entries, update the small per-client memory, and generate HPs for the next round (client/server learning rates, momenta, schedulers, batch size, local epochs, dropout, and FedProx $\mu$). Updated HPs are written to a lightweight store read by clients at the start of the next round. Training never waits for the agent.

*Overlap and latency.* Synchronous designs incur $T_{\text{train}} + T_{\text{agent}}$; our non-blocking design approaches $\max(T_{\text{train}}, T_{\text{agent}})$ because agent calls run in parallel with GPU computation. Occasional slow responses defer HP changes to the following round. Under this scheme, we observed up to a 39% reduction in per-client wall-clock time.

*Fault tolerance.* If an agent request fails or times out, training continues using the most recent valid HPs or safe defaults. All agent outputs are range-checked, and invalid fields are clamped or discarded. Only scalar summaries are exchanged, maintaining the privacy guarantees of split learning. (See Appendix A. 2 for pseudocode.)

## 3.3 PHASE 3: SERVER AGGREGATION

At the end of each round, aggregation proceeds in a SplitFed-consistent manner. During the round, each selected client $i \in S_t$ updates its client-side parameters $w_{c,i}$ by approximately minimizing a FedProx local objective

$$\min_w \; f_i(w) \; + \; \frac{\mu}{2}\, \|w - w_c^{(t)}\|^2,$$

where $f_i$ is the client's empirical loss and $\mu$ is the proximal coefficient proposed by the HP agent. This proximal term regularizes local steps toward the current global client-side weights $w_c^{(t)}$, mitigating drift that is amplified by non-IID data, personalized hyperparameters, and cluster-specific split points. In parallel, the server updates its server-side parameters $w_s$ centrally using the streamed activations and their cut gradients; no client-side averaging is involved for $w_s$.

After local updates complete, the server aggregates only the *client-side* submodel weights by data-weighted averaging Li et al. (2020b):

$$w_c^{(t+1)} \; = \; \sum_{i \in S_t} \frac{n_i}{\sum_{j \in S_t} n_j} \; w_{c,i}^{(t+1)},$$

where $n_i$ is client $i$'s local sample count. The server-side parameters $w_s^{(t+1)}$ are those obtained from the server's in-round optimization. Finally, the pair $\left(w_c^{(t+1)},\, w_s^{(t+1)}\right)$ together with the next-round hyperparameters $(\eta_c^{(t+1)}, \eta_s^{(t+1)}, \mu^{(t+1)})$ are broadcast to the next cohort of clients. This separation—local FedProx on the client-side submodel, centralized updates on the server-side submodel, and FedAvg-style aggregation only for $w_c$—matches the split-learning execution while preserving stability under heterogeneity.

## 3.4 DUAL-AGENT OPTIMIZATION ENGINE

FedAgent-HPO employs two asynchronous LLM-driven components—the *HP Agent* and the *Analyzer*—that operate on compact, structured feedback rather than raw training signals. For each client we maintain a small agent-side state with a sliding window ($W$=5) of round-level summaries: scalar loss and accuracy, a simple step-time statistic, and the hyperparameters used in that round. The window length follows common practice in LLM agent memory systems, where fixed-size buffers (3–5 recent interactions) capture salient trends without saturating context Zhang et al. (2025). To improve data efficiency early in training, agents also read peer-informed priors in the form of cluster-level aggregates (e.g., mean/best learning rates and interquartile ranges of metrics) derived from clients with similar resource profiles. Unlike population-level heuristics such as Fed-Pop Chen et al. (2025a), these summaries are resource-aware and remain aggregate-only.

At the end of a round, the HP Agent fetch the client's recent summary window together with the cluster aggregates and proposes a configuration for the next round. The configuration spans continuous parameters (learning rate, weight decay, dropout), bounded integers (local epochs, batch size subject to device capacity), and categorical choices (optimizer, scheduler). Running alongside, the Analyzer maintains a search-space state per client, widening or narrowing

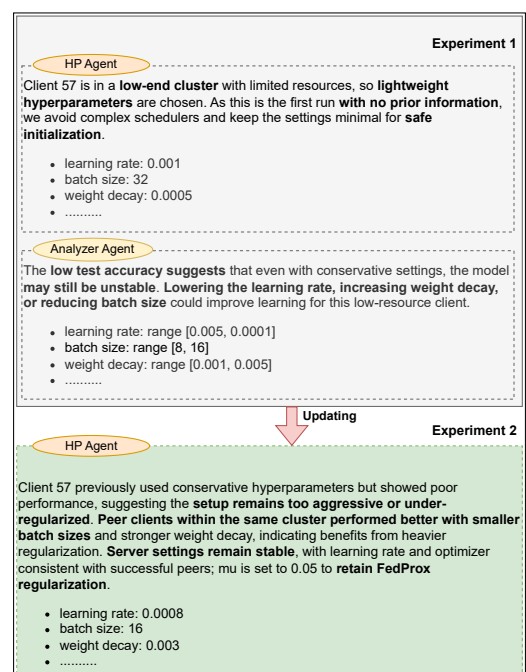

Figure 2: Dual-agent reasoning on client summaries. The HP Agent proposes next-round hyperparameters using a small per-client memory and cluster-level peers; the Analyzer maintains per-client constraints on the search space. Proposals are validated before commit to ensure adherence to Analyzer ranges and allowed sets.

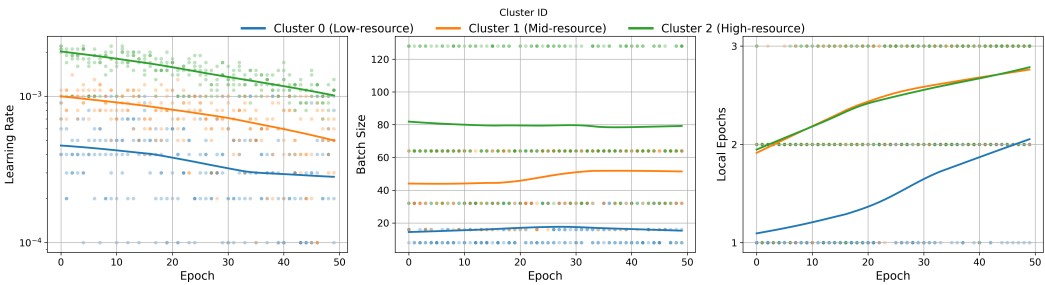

Figure 3: Evolution of key hyperparameters (learning rate, batch size, local epochs) over training epochs across client clusters.

continuous ranges based on observed trends and
removing poorly performing categorical options when justified by recent evidence.

To ensure consistency and safety, all HP proposals are passed through a validator that enforces the Analyzer's constraints before commit. Continuous values are clamped to the current per-client ranges; categorical fields must belong to the allowed set; bounded integers are projected to the feasible interval and device-aware caps. Malformed or schema-violating outputs are deterministically repaired when possible; otherwise the system retains the last valid value. If an agent call fails or times out, training proceeds and a deterministic fallback chain selects the next configuration without blocking: *last-good hp → peer-informed prior* (cluster aggregate) *→ safe defaults*. The shared queue is non-blocking with bounded capacity; if it fills, the oldest metrics record is dropped (never a computed HP decision). This design yields interpretable, per-client optimization paths while remaining robust under heterogeneous conditions *(Prompt templates and example JSON outputs are provided in Appendix A.3).*

To empirically validate the tuning behavior of our LLM-based HP Agent, we visualize in Figure 3 the evolution of selected hyperparameters over training epochs across different client clusters. These clusters correspond to heterogeneous resource profiles (low, mid, and high). The figure presents smoothed per-epoch trajectories for learning rate, batch size, and local epochs. We observe that the HP Agent adapts its tuning strategy based on client context: low-resource clients converge rapidly to more conservative configurations, whereas high-resource clients engage in broader hyperparameter exploration before gradually narrowing the search space. This demonstrates that our agent performs adaptive, resource-aware hyperparameter optimization rather than relying on static schedules or heuristic rules.

### 3.5 COST ANALYSIS

We compare the computational and communication costs of our agent-based HPO with conventional wrapper-based baselines and tuning-while-training methods.

Traditional methods such as Random Search (RS) Bergstra & Bengio (2012) and Successive Halving (SHA) Jamieson & Talwalkar (2016) require multiple full FL runs, each with a different configuration. Their total cost scales as $\mathcal{O}(nET)$ time and $\mathcal{O}(nEYS)$ communication, where $n$ is the number of configurations, $E$ the number of rounds, $Y$ participating clients, $S$ the update size, and $T$ the per-round runtime. SHA reduces cost via early stopping Li & Jamieson (2018), but both remain linear in $n$ and thus expensive in federated settings.

| Method | Time Complexity | Communication Complexity |
|--------|-----------------|--------------------------|
| RS | $nET$ | $nEYS$ |
| SHA | $\frac{nET}{\log_2 n}$ | $\frac{nEYS}{\log_2 n}$ |
| LLM-Agent | $E(T+kL)$ | $EYS + EYP_{\text{API}}$ |

Table 1: Asymptotic complexity of HPO strategies in federated learning.

In our method, HPO is embedded within a single continuous run. As shown in Table 1, the dominant cost remains $\mathcal{O}(ET)$ from training, while agent reasoning adds only $k$ lightweight calls per round (here $k=2$), giving $\mathcal{O}(E(T+kL))$ total time. Communication overhead from agent queries is negligible since $P_{\text{API}} \ll S$, yielding $\mathcal{O}(EYS+EYP_{\text{API}})$ overall. Because agent calls run asynchronously on CPU while GPUs continue training, the added latency is minimal in practice.

SplitFed further lowers communication. As summarized in Table 2, standard FL requires $2|W|$ communication per client per round, whereas SplitFed transmits only $pK|W|$ where $p/K<1$ reflects the client-side model fraction. Aggregation costs shrink proportionally. Together, partial-model communication and in-line agent tuning provide a computationally and communication-efficient solution for large-scale heterogeneous FL.

| Method | Comms. per client | Total comms. | Total model time |
|---|---|---|---|
| FL | $2|\mathbf{W}|$ | $2K|\mathbf{W}|$ | $T + \frac{|\mathbf{W}|}{R} + T_{\text{fedavg}}$ |
| Split FL | $\frac{p}{K}|\mathbf{W}|$ | $pq|\mathbf{W}|$ | $T + \frac{p}{KR} + \frac{T_{\text{fedavg}}}{2}$ |

Table 2: Cost per epoch for FL vs. Split FL. Split FL reduces cost when $\frac{p}{K}|\mathbf{W}| < 2|\mathbf{W}|$ and $\frac{T_{\text{fedavg}}}{2} < T_{\text{fedavg}}$.

## 4 EXPERIMENTS AND ANALYSES

### 4.1 BENCHMARK EXPERIMENTS

**Dataset and Model Description** We evaluate FedAgent-HPO against state-of-the-art HPO baselines on standard federated benchmarks Khodak et al. (2021), including cross-silo experiments, LLM backend comparisons, and ablations. All experiments are conducted on NVIDIA A100 GPUs.

We use three widely adopted FL datasets spanning vision and language: (1) *CIFAR-10* Krizhevsky et al. (2012) for image classification, (2) *FEMNIST* Caldas et al. (2018) for handwritten character recognition, and (3) *Shakespeare* Caldas et al. (2018) for next-character modeling. Following standard practice, we consider both IID and non-IID settings: FEMNIST clients correspond to individual writers, Shakespeare clients to characters, and CIFAR-10 non-IID partitions follow a Dirichlet prior $\text{Dir}(\alpha)$ Zhu et al. (2021); Lin et al. (2020). We simulate 500 clients for CIFAR-10, 3550 for FEMNIST, and 1129 for Shakespeare, using ResNet18 He et al. (2016) and CharLSTM Kim et al. (2016) for vision and language tasks, respectively.

**Results and Discussion**

Table 4 reports final global accuracies across three benchmarks. The LLM Agent (gpt-4o-mini), with and without SplitFed, consistently outperforms all baselines: on CIFAR-10 (IID) Vanilla FL is +4.8% over the best baseline and SplitFed is +4.0%; under non-IID, gains widen to +7.7–8.8% (Vanilla) and +7.0–8.5% (SplitFed). On FEMNIST, SplitFed is +2.0% under user-partitioned non-IID and matches baselines in IID. On Shakespeare (non-IID), SplitFed shows the largest relative gain at +4.9%. Two patterns emerge: (i) the IID→non-IID drop is only 2–3% for our methods vs. 5–7% for conventional approaches, indicating stronger robustness; (ii) Vanilla FL is slightly better on vision tasks, while SplitFed leads on the sequential language task, likely because centralizing deeper layers helps aggregate longer-range context—consistent with observations in heterogeneity-aware split learning Zhang et al. (2022).

### 4.2 CROSS-SILO FL VALIDATION

To test FedAgent-HPO in practical settings, we evaluate it on cross-silo federated learning, where each client is a large domain-specific silo. Unlike cross-device FL, these scenarios have fewer clients but higher domain heterogeneity, making hyperparameter tuning crucial. Hyperparameters tuned on cross-device benchmarks often fail to generalize due to differences in data and model characteristics Kuo et al. (2023), which motivates this evaluation.

We use two popular cross-silo datasets commonly employed in domain generalization and federated learning research: PACS Li et al. (2017) contains four visual domains (Art Painting, Cartoon, Photo, and Sketch) spanning seven object categories, while OfficeHome Venkateswara et al. (2017) consists of 65 classes across four styles: Art, Clipart, Product, and Real World. Following standard cross-silo protocols Li et al. (2020a); Qi et al. (2023), each domain is treated as a silo. For

Table 3: Performance on cross-silo FL datasets.

| Algorithm | PACS | OfficeHome |
|---|---|---|
| SHA | 76.53 | 57.64 |
| FedEx | 80.61 | 58.40 |
| FedPop | 85.37 | 62.76 |
| **FedAgent-HPO** | **89.72** | **74.20** |

Table 4: Accuracy of different hyperparameter tuning methods on CIFAR-10, FEMNIST, and Shakespeare under IID and non-IID settings (*Dir=1.0 and Dir=0.5 indicate different levels of data heterogeneity*). Top scores in bold.

| Method | CIFAR-10 | | | FEMNIST | | Shakespeare | |
|---|---|---|---|---|---|---|---|
| | IID | NIID (Dir-1.0) | NIID (Dir-0.5) | IID | NIID | IID | NIID |
| RS | 69.04 | 63.47 | 62.88 | 82.86 | 79.06 | 33.76 | 32.67 |
| RS + FedEx | 67.91 | 64.34 | 63.22 | 82.84 | 82.14 | 42.68 | 44.28 |
| RS + FedPop | 71.18 | 68.25 | 67.01 | 84.33 | 83.21 | 44.30 | 47.28 |
| SHA | 78.57 | 70.37 | 68.65 | 83.81 | 80.62 | 52.23 | 51.68 |
| SHA + FedEx | 79.83 | 72.02 | 69.69 | 81.19 | 82.76 | 51.79 | 51.26 |
| SHA + FedPop | 81.47 | 76.42 | 74.88 | 84.33 | 83.26 | 53.48 | 53.07 |
| **FedAgent-HPO (w/o SplitFed)** | **86.28** | **84.13** | **83.67** | **85.23** | **83.89** | **53.64** | **53.94** |
| **FedAgent-HPO (with SplitFed)** | **85.52** | **83.46** | **83.37** | **86.21** | **85.26** | **57.83** | **58.01** |

evaluation, we hold out Photo as the test silo in PACS and Real World in OfficeHome. All experiments use a ResNet-18 backbone, consistent with prior cross-silo FL studies.

Table 3 shows that FedAgent-HPO achieves consistently higher accuracy across both cross-silo benchmarks, achieving an improvement of 4.3 percentage points on PACS and 11.4 percentage points on OfficeHome. The OfficeHome result ($\sim$74%) is particularly significant, exceeding the 62% average reported in the FedPop study. This margin aligns with recent findings that per-silo hyperparameter adaptation can yield double-digit gains under severe domain shifts Park et al. (2023). It also reflects the well-documented variance of OfficeHome, where carefully tuned ResNet-18 models regularly range from the mid-60s to mid-70s depending on optimization settings, and FedPop itself reports standard deviations as high as 7–15%.

## 4.3 IMPACT OF LLM MODEL SELECTION

Table 5 highlights that the LLM agent choice impacts both accuracy and the overall dynamics of federated HPO. For clarity, *API Fail. (%)* measures timeout/schema-invalid rates, *Avg. Tokens* counts tokens per response, and *Avg. Time (s)* is end-to-end latency per call. Llama 4 achieves slightly lower accuracy than GPT 4o Mini but offers near-perfect reliability, showing that stable API behavior can be as valuable as raw model quality by avoiding stale hyperparameters and maintaining consistent optimization. Grok-3-mini produces overly long outputs, inflating $T_{\text{agent}}$ and reducing the gains of asynchronous scheduling, high-

Table 5: Performance and cost comparison of LLM agents on CIFAR-10.

| LLM Agent | Acc. (%) | API Fail. (%) | Avg. Tokens | Cost / 1K calls ($) |
|---|---|---|---|---|
| GPT-4o-mini | 83.46 | 1.2 | 628 | 0.24 |
| Llama 4 Maverick | 82.83 | 0.2 | 621 | 0.23 |
| Grok-3-mini | 79.43 | 0.4 | 3444 | 1.29 |
| Gemma 2 (9B) | 82.09 | 12.93 | 223 | 0.00 |

Using OpenRouter pricing for GPT-4o-mini, average 628-token agent call across 500 clients and 100 rounds results in a total cost of about $12.

lighting that concise reasoning better aligns with $\max(T_{\text{train}}, T_{\text{agent}})$ execution. Gemma 2 illustrates the trade-off of free models: its higher failure rate forces frequent reuse of $\mathcal{H}_i^t$, shrinking the search space and reducing adaptivity. While cost-efficient, this underscores how model reliability directly shapes optimization dynamics. Overall, effective federated HPO agents must balance reasoning quality, latency, and service stability, as even small variances in output or failure rates propagate through the asynchronous pipeline and affect convergence and personalization.

## 4.4 ABLATION STUDIES

To evaluate the impact of scheduling strategy, we compare monolithic scheduling, where LLM-based reasoning and model training occur sequentially, with an asynchronous agent–GPU pipeline that executes them in parallel.

Figure 4 quantifies the effect of decoupling agent reasoning from GPU training.

The asynchronous pipeline reduces per-epoch time by overlapping LLM computation with model updates. On CharLSTM (Shakespeare), we observe a $\sim25\%$ reduction, while ResNet-18 (CIFAR-10) achieves $\sim39\%$. These results align with our latency model, where asynchronous execution reduces round time from $T_{\text{train}} + T_{\text{agent}}$ to $\max(T_{\text{train}}, T_{\text{agent}})$, confirming that when agent latency approaches training cost, parallelization amortizes overhead and generalizes across architectures without additional tuning.

Table 6 highlights that both historical context and peer information contribute significantly to stable hyperparameter optimization. Removing historical client data leads to moderate performance degradation, indicating that short-term memory drives per-client adaptation. Excluding peer information causes a larger drop, showing that cross-client signals are crucial for early-round generalization when local history is sparse. Disabling regularization further amplifies client drift under non-IID conditions, underscoring the importance of proximal constraints in personalized settings. The non-reasoning variant incurs the steepest degradation, demonstrating that eliminating all contextual reasoning establishes a clear lower bound for agentic optimization. The results here are cumulative: removing peer information builds on the loss of historical context, no regularization excludes both, and the non-reasoning variant combines all removals, showing that FedAgent-HPO's gains come from the combined effect of memory, peer reasoning, and regularized aggregation rather than any single factor.

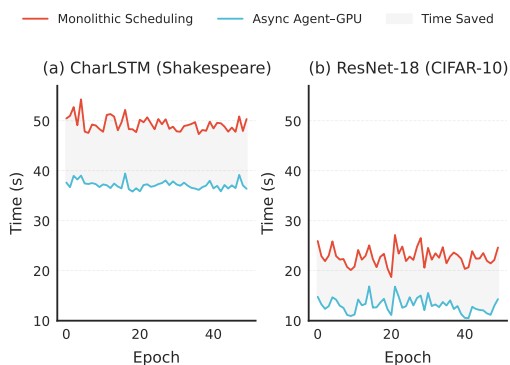

Figure 4: Comparison of monolithic scheduling and asynchronous agent–GPU pipeline. Shaded area shows time saved by overlapping LLM and training.

Table 6: Ablation study with accuracy drops (in parentheses) relative to the baseline LLM Agent.

| | CIFAR-10 | | |
|---|---|---|---|
| **Ablation Variant** | **IID** | **NIID (Dir 1.0)** | **NIID (Dir 0.5)** |
| **FedAgent-HPO (Baseline)** | **85.52** | **83.46** | **83.37** |
| Without Historical Context | 84.50 ($\downarrow$1.02) | 83.12 ($\downarrow$0.34) | 82.77 ($\downarrow$0.60) |
| Without Peer Information | 83.13 ($\downarrow$2.39) | 81.33 ($\downarrow$2.13) | 80.66 ($\downarrow$2.71) |
| No Regularization | 82.78 ($\downarrow$2.74) | 81.03 ($\downarrow$2.43) | 80.32 ($\downarrow$3.05) |
| Non-Reasoning LLM Agent | 81.23 ($\downarrow$4.29) | 80.06 ($\downarrow$3.40) | 79.57 ($\downarrow$3.80) |

## 5 CONCLUSION AND FUTURE WORK

FedAgent-HPO establishes a dual-agent LLM paradigm for personalized hyperparameter optimization in federated learning, combining asynchronous CPU–GPU scheduling with SplitFed to enable per-client tuning with efficient training dynamics, reduced communication cost, and improved accuracy under both IID and non-IID settings. Its interpretable, client-aware trajectories demonstrate adaptive tuning in heterogeneous environments. The reliance on external LLMs remains a challenge, motivating future work on lightweight on-device agents to make the framework more efficient and deployable in resource-constrained edge environments.[1]

---

[1]A large-language-model assistant was used only for minor copy-editing and for suggesting related work. All technical ideas, equations, algorithms, and analyses were developed and verified by the authors; all cited references were manually checked and included only after reading.

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

# A  APPENDIX

This appendix collects additional details that complement Sec. 3 (Method) and Sec. 5 (Results).

**A.1 Hyperparameter Search Space** lists all client/server knobs, their types (choice vs. numeric), and initial ranges or allowed sets.

**A.2 Algorithms** gives pseudocode for SplitFed with resource-aware clustering and the asynchronous agent–trainer design; decisions are applied at the start of a client's next participation round.

**A.3 Agent Prompts and Output** includes the exact prompt templates and output JSON responses for the Analyzer and HP agents.

## A.1  HYPERPARAMETER SEARCH SPACE

In this subsection, we define the hyperparameter search space used in FedAgent-HPO. The ranges were selected based on standard federated learning practices and preliminary experiments to ensure stable convergence across heterogeneous clients. Both client- and server-side parameters, as well as the global FedProx coefficient, are tuned dynamically during training.

| Hyperparameter | Type | Range / Values | Initial |
|---|---|:---:|:---:|
| **Client-side** | | | |
| Learning rate | float | [0.0001, 0.01] | 0.001 |
| Weight decay | float | $[10^{-6}, 0.001]$ | $5 \times 10^{-5}$ |
| Momentum (SGD) | float | [0.8, 0.99] | 0.9 |
| Optimizer | choice | {AdamW, SGD, Adam} | AdamW |
| Scheduler | choice | {CosineAnnealingLR, StepLR, None} | CosineAnnealingLR |
| Local epochs | int | [1, 3] | 2 |
| Batch size | choice | {8, 16, 32, 64, 128} | 64 |
| Dropout rate | float | [0.0, 0.5] | 0.1 |
| **Server-side** | | | |
| Learning rate | float | [0.0001, 0.01] | 0.001 |
| Momentum (SGD) | float | [0.8, 0.99] | 0.9 |
| Optimizer | choice | {AdamW, SGD, Adam} | AdamW |
| Scheduler | choice | {CosineAnnealingLR, StepLR, None} | None |
| **Global** | | | |
| $\mu$ | float | [0.001, 0.1] | 0.01 |

Table 7: Hyperparameter search space for client, server, and global parameters used in FedAgent-HPO.

## A.2  ALGORITHMS

This appendix gives the precise execution flow for Sec. 3: (i) SplitFed with resource-aware clustering (Alg. 1), and (ii) the asynchronous agent loop coordinated with GPU training (Alg. 2).

**Signals and privacy.** The agent consumes only compact scalar summaries (loss, accuracy, step time, and the HPs *used* in the last round). No activations, gradients, or raw data leave the client.

**Execution model.** Training (GPU) and reasoning (CPU) run concurrently. A non-blocking metrics queue connects the threads; if the queue is full, the *oldest metrics record* is dropped and training continues. Decisions are written to a per-client store (NextHP[i]) and never overwrite a computed decision in flight.

**Clustering and split.** Resource profiling (memory, bandwidth, short warmup step time) produces features for clustering; each cluster selects a split point that fits typical bandwidth/latency.

**Validation and fallbacks.** Proposed HPs are projected onto each client's feasible set (range clamping, integer bounds, allowed categories; batch size capped by memory at the chosen split). On error or invalid output, we use a deterministic fallback: last-good → cluster prior → safe defaults.

---

**Algorithm 1:** SplitFed with resource-aware clustering (profiling $\rightarrow$ clustering $\rightarrow$ split) (Sec. 3.1)

---

**Input:** Initial weights $w_c^{(0)}, w_s^{(0)}$; number of clusters $K$; candidate cut layers $\mathcal{Z}$; optional re-cluster cadence $R$

**Result:** Cluster assignments, per-cluster split points, and trained $(w_c^{(E)}, w_s^{(E)})$

---

1 **Step 1: Lightweight profiling (one-time)**;

2 **foreach** *client* $i$ **do**

3     Measure a small device/link profile: available memory, GPU type/availability, uplink/downlink bandwidth, and a short warmup step-time;

4     Build a feature vector $\phi_i$ from these measurements;

5 **Step 2: Clustering and split selection**;

6 Cluster clients $\{1, \ldots, N\}$ into $\{C_1, \ldots, C_K\}$ using the features $\{\phi_i\}$;

7 **foreach** *cluster* $C_k$ **do**

8     Choose a split point $z_k \in \mathcal{Z}$ that fits typical bandwidth/latency in $C_k$ (guided by measured activation size $|z_k|$ and per-batch traffic $2b|z_k|$);

9 Assign each client $i$ to its cluster $c(i)$ and record its split $z_{c(i)}$;

10 **Step 3: Training rounds (uses the split; HP logic handled in Algorithm 2)**;

11 **for** *round* $t = 0, 1, \ldots, E - 1$ **do**

12     **if** $R > 0$ *and* $t > 0$ *and* $t \bmod R = 0$ **then**

        `// Optional drift handling`

13         Refresh recent bandwidth/step-time samples; re-cluster and update splits if needed;

14     Sample participating clients $S_t$ and broadcast $(w_c^{(t)}, w_s^{(t)})$;

15     **foreach** $i \in S_t$ **do**

16         Set the cut to $z_{c(i)}$; run local forward passes to produce $z$; send $z$ to server and receive $\partial\ell/\partial z$;

17         Backprop through the client-side model and update local $w_{c,i}$ (e.g., FedProx locally);

18         Send $\Delta w_{c,i}$ to the server if parameter sync is enabled;

19     On the server: update $w_s$ using streamed $z$ and $\partial\ell/\partial z$; aggregate client-side weights to obtain $w_c^{(t+1)}$; broadcast $(w_c^{(t+1)}, w_s^{(t+1)})$;

---

---

**Algorithm 2:** Asynchronous execution: GPU training thread + CPU agent thread (Sec. 3.2)

---

**Input:** Non-blocking metrics queue $Q$ (capacity $M$); per-client store `NextHP[]` (default = last-good or safe default)

**Result:** Per-client hyperparameters applied *at the start* of the next participation round

---

**1 Thread 1 — GPU training (runs on clients/server)**;

**2 for** *round* $t = 0, 1, \ldots, E-1$ **do**

**3**     Sample clients $S_t$ and broadcast $(w_c^{(t)}, w_s^{(t)})$;

**4**     **foreach** *client* $i \in S_t$ **do**

       `// Apply next-round decision, if available`

**5**        **if** *NextHP[i] exists* **then**

**6**           set current HPs $\leftarrow$ `NextHP[i]`

**7**        **else**

**8**           keep last-good HPs

**9**        Run split training with GPU: forward on client, send $z$, receive $\partial\ell/\partial z$, backprop on client, update local $w_{c,i}$;

**10**        Compute tiny summaries: `loss`, `acc`, `step_time`, and record the HPs *used* this round;

       `// Produce metrics for the agent without blocking training`

**11**        **if** *$Q$ is full* **then**

**12**           drop *oldest* metrics record in $Q$

**13**        Push $\langle i, t, \texttt{loss}, \texttt{acc}, \texttt{step\_time}, \texttt{HPs\_used}\rangle$ into $Q$;

**14**     Server updates $w_s$ from streamed $z$ and $\partial\ell/\partial z$; aggregate $w_c^{(t+1)}$ and broadcast;

**15 Thread B — CPU agent (runs in parallel)**;

**16 while** *training active* **do**

**17**     **if** *$Q$ is empty* **then**

**18**        **continue** ;                             `// no waiting`

**19**     Pop newest record $\langle i, t, \texttt{loss}, \texttt{acc}, \texttt{step\_time}, \texttt{HPs\_used}\rangle$ from $Q$;

**20**     Update a short per-client history (e.g., last 5 entries);

     `// Analyzer refines what is allowed; HP Agent proposes within context`

**21**     Refine feasible ranges/options for client $i$ using history + light cluster summary;

**22**     Propose next HPs for client $i$ based on that context;

     `// Hard validation so decisions are safe on device/split`

**23**     Clamp to ranges, cap batch size by memory at the client's split, and drop disallowed options;

**24**     **if** *proposal invalid or agent error* **then**

**25**        use fallback: last-good $\rightarrow$ cluster prior $\rightarrow$ safe defaults

     `// Commit for next time client i is scheduled`

**26**     Set `NextHP[i]` $\leftarrow$ validated HPs (effective from the start of round $t+1$ for client $i$);

---

## A.3   AGENT PROMPTS AND OUTPUT

We include the exact prompt templates used by the HP Agent and the Analyzer Agent. Placeholders (e.g., `{{client_id}}`, `{history_str})`, `{task_description})`, etc. are populated at runtime with scalar summaries and constraint blocks.

Both agents are instructed to return *strict JSON* that our validator checks before use; any invalid output triggers a deterministic fallback (last-good $\rightarrow$ cluster prior $\rightarrow$ safe defaults). The HP Agent produce a JSON object with keys `reasoning` and `hps` (containing `client`, `server`, and `mu`); the Analyzer emits `reasoning` and `actions` (range clamps or allowed-set edits).

Listing 1: HP Agent system prompt

```
You are an expert ML engineer and strategist. Your task is to act as a
    methodical scientist, analyzing all available data to suggest the
    optimal set of hyperparameters for both the client and the server.

YOUR THOUGHT PROCESS:
1.Analyze the Task: First, identify the task (`{task_description}`). Your
     strategy for text prediction should be different from image
    classification.
2.Review History: Examine the client's own history and peer history. Is
    there a pattern? Are high learning rates consistently failing? Is
    dropout helping?
3.Consult Guidance: Read the expert guidance below and prioritize your
    choices based on it.
4.Formulate Reasoning: In the "reasoning" key, explain *why* you are
    choosing each value, referencing the history and guidance.
5.Construct Output: Provide a single, valid JSON object according to the
    format, ensuring all values adhere to the STRICT CONSTRAINTS.
-----------------------------------------
OVERALL CONTEXT:
- Client ID: {client_id}
- Model: {model_name} on {dataset_name}
- Task: {task_description}
- Federated Scheme: SplitFed with FedProx regularization (controlled by `
    mu`).
-----------------------------------------
CLIENT-SIDE CONTEXT & INSTRUCTIONS
- Client Capacity:{_get_cluster_capacity_string(cluster_id)}. Low-
    resource clients may need smaller `batch_size`, fewer `local_epochs`,
    or lower `learning_rate`.
- Client's Own History:{history_str}
- Peer History:{peer_history_str}
- Last Round's Analysis: {analysis_str}
-----------------------------------------
STRICT CONSTRAINTS - YOU MUST FOLLOW THESE EXACTLY:
Client Parameters:{client_constraints_str}
Server Parameters:{server_constraints_str}
Global Parameter:{mu_constraint}
-----------------------------------------
CRITICAL RULES:
1. DO NOT suggest values outside the allowed ranges/choices
2. DO NOT be creative with batch sizes** - only use the exact values
    listed
3. DO NOT suggest optimizers not in the list
4. STICK TO THE CONSTRAINTS - they exist for a reason
-----------------------------------------
OUTPUT FORMAT & INSTRUCTIONS
- Return a single, valid JSON object with "reasoning" and "hps" keys.
- The "hps" object must contain "client", "server", and "mu" keys.
-----------------------------------------
FINAL INSTRUCTION:
-Output MUST be a single valid JSON object exactly matching the requested
     keys.
-Do not include any explanations outside the JSON.
-Do not use Markdown or code fences.
-Do not include any introductory or concluding text.
-----------------------------------------
YOUR RESPONSE MUST BE PURE JSON.
```

Listing 2: Analyzer Agent system prompt

```
You are an expert Machine learning HPO analysis agent.
----------------------------------------
TASK:
1. Analyze the client's performance and provide a list of actions to *
    refine the hyperparameter search space for future rounds*.
2. Your actions should be strategic, aiming to guide the hyperparameter
    search towards better solutions over time.
3. Return a JSON object with "reasoning" and "actions" keys.
----------------------------------------
CONTEXT:
- Client: {client_id} (Epoch: {global_epoch + 1}, Capacity: {
    _get_cluster_capacity_string(cluster_id)})
- Task: {task_description}
- HPs Used: {json.dumps(current_hps)}
- Result: {results_str}
----------------------------------------
STRATEGIC GUIDANCE FOR ANALYSIS:
- Your Goal: Your job is to act like a researcher guiding an experiment.
    You must prune the search space to eliminate bad regions and focus on
     promising ones.
- If Overfitting Persists: If multiple rounds show a large gap between
    train and test accuracy, the current search space is too aggressive.
    Your actions should make the search space more conservative. For
    example: {"param": "learning_rate", "key": "max", "value": 0.001} or
    {"param": "dropout_rate", "key": "min", "value": 0.3}.
- If Performance Stagnates: If accuracy is stuck, the search space might
    be too narrow. Consider actions that slightly expand a key parameter
    range to encourage exploration.
----------------------------------------
CRITICAL INSTRUCTIONS FOR ACTIONS:
1.You MUST use the correct modification key for each parameter based on
    its type.
    - For CHOICE parameters (use "key": "values"):
    - Client: {client_choice_params}
    - Server: {server_choice_params}
    - For NUMERICAL parameters (use "key": "min" or "max"):
        -Client: {client_numerical_params}
        -Server: {server_numerical_params}

2.Parameter Names: Use ONLY the exact names from the lists above.
    -    For client parameters, set "target": "client_hps".
    -    For server parameters, set "target": "server_hps".

3.Value Formatting: The "value" field format MUST match the key.
    -If "key" is "values", then "value" MUST be a LIST (e.g., [16, 32]).
    -If "key" is "min" or "max", then "value" MUST be a single NUMBER (e.
    g., 0.005).
----------------------------------------
FINAL INSTRUCTION:
-Output MUST be a single valid JSON object exactly matching the requested
     keys.
-Do not include any explanations outside the JSON.
-Do not use Markdown or code fences.
-Do not include any introductory or concluding text.
----------------------------------------
YOUR RESPONSE MUST BE PURE JSON.
```

Listing 3: HP Agent Response

```
<<< RESPONSE FROM HP AGENT (Client 327):
----------------------------------------
Reasoning: Given that this is the first round for the client and
    considering they belong to a low-resource cluster, I will proceed
    with conservative hyperparameter choices. For learning_rate, I will
    set it to 0.001 to balance stability and convergence speed. The
    weight_decay will be set to 0.0001 to incorporate regularization
    without overly penalizing the model. A momentum of 0.9 is a standard
    choice that helps in accelerating SGD in the relevant directions. The
     optimizer will be 'Adam' as it's generally effective for image
    classification tasks. Since the clients are low-resource, I will
    choose a batch_size of 16, which allows them to use their resources
    efficiently without running into memory issues, and I will set
    local_epochs to 2 for sufficient training. Lastly, the dropout_rate
    will be set to 0.2, as it helps in regularization. For the server, I
    will mirror the learning_rate and momentum used in the client, opting
     for 'Adam' again for consistency, and use 'None' for the scheduler
    since it's only the first round. Lastly, I suggest mu set to 0.01,
    which balances the federated learning's local divergence and global
    convergence.
----------------------------------------
 "hps": {
        "client": {
            "learning_rate": 0.001,
            "weight_decay": 0.0001,
            "momentum": 0.9,
            "optimizer": "Adam",
            "scheduler": "None",
            "local_epochs": 3,
            "batch_size": 16,
            "dropout_rate": 0.2
        },
        "server": {
            "learning_rate": 0.001,
            "momentum": 0.9,
            "optimizer": "Adam",
            "scheduler": "None"
        },
        "mu": 0.01
    }
}
----------------------------------------
....
----------------------------------------
<<< RESPONSE FROM HP AGENT (Client 118):
----------------------------------------
Reasoning:
For this initial round, I will suggest hyperparameters that are
    reasonable based on client and peer  history. Since Client 276
    achieved a decent accuracy of 70.86% with a learning rate of 0.0001,
    I will adopt the same learning rate for Client 118 to ensure a solid
    start and avoid potential overfitting issues at this early stage
    .......
----------------------------------------
    "hps": {
        "client": {
            "learning_rate": 0.0001,
            ...
        "mu": 0.01
    }
}
------------------------------------------------------------
```

Listing 4: Analyzer Agent Response

```
......
Client 327: Train Acc 64.86%, Test Acc 54.96%
------------------------------------
<<< RESPONSE FROM ANALYZER AGENT (Client 327):
------------------------------------
Reasoning:
The achieved accuracy of 54.96% is relatively low, indicating potential
    issues with model training or hyperparameter settings. Decreasing the
     learning rate and increasing weight decay could improve the training
     stability for this low-resource client. Additionally, exploring
    smaller batch sizes will be beneficial for the training process.
------------------------------------
Actions: [

        {
            "param": "batch_size",
            "key": "values",
            "value": [8, 16],
            "target": "client_hps"
        },
        {
            "param": "optimizer",
            "key": "values",
            "value": ["Adam", "SGD"],
            "target": "client_hps"
        }
        ....
        ....
    ]
}
```

