# OpenReview forum: "FedAgent-HPO: Agentic Hyperparameter Optimization for Personalized Federated Learning"
_ICLR.cc/2026/Conference — ICLR 2026 Conference Withdrawn Submission_

### Official Review · Reviewer_YCVK · 2025-10-14

**Soundness:** 2
**Presentation:** 2
**Contribution:** 2
**Rating:** 4
**Confidence:** 3

**Summary:**

This paper proposes FedAgent-HPO, a dual-agent framework for personalized hyperparameter optimization (HPO) in FL. It combines a Hyperparameter Agent and an Analyzer Agent. The system dynamically adapts per-client hyperparameters online using summaries, resource-aware clustering, and asynchronous CPU-GPU scheduling with SplitFed architecture. The proposed method demonstrates promising accuracy and efficiency across multiple FL benchmarks.

**Strengths:**

1. The problem setting is interesting and remains under-explored.
2. The writting is clear.
3. The experimental results are promising.

**Weaknesses:**

1. The communication costs of the FedSplit seems to be ignored from Table 1.
2. The definition and simulation of the settings for resource heterogeneity is unclear. Also, there is no details/hyperparameters about how the clustering is performed.
3. There is no analysis/visualization about the tuning process. It is unclear how the LLM-Agent select/tune the hyperparameters.
4. There is no variation reported in the model performance. The authors also removed the performance variation from the previous work (FedPop), which makes the robustness of the tuning results questionable.

**Questions:**

Please refer to the Weaknesses section.

---

### Official Review · Reviewer_M62M · 2025-10-17

**Soundness:** 2
**Presentation:** 3
**Contribution:** 2
**Rating:** 4
**Confidence:** 2

**Summary:**

This paper introduces FedAgent-HPO, a framework that utilizes Large Language Model agents for personalized Hyperparameter Optimization (HPO) in FL. Generally, the idea is to use an online, personalized approach where LLM agents dynamically adjust hyperparameters during a single training run. Such an approach may be able to adapt to heterogeneity and resource constraints in real-time.

**Strengths:**

1. The paper argues that using LLM-based reasoning for personalized HPO in FL is efficient. This is not a commonly used approach, so is unique in some sense.

2. The framework appears to demonstrate substantial accuracy improvements over strong baselines across diverse benchmarks.

3. This approach may provide natural language reasoning for hyperparameter choices.

**Weaknesses:**

1. There are no rigorous convergence bounds, but it is difficult to imagine how something like this may be proposed, based on the framework.

2. In what case might this framework fail? Whenever LLM agents are involved in hyperparameter optimization, wouldn't this be a very strong possibility? Have the failure modes been analyzed?

**Questions:**

Please see weaknesses. Not much questions to add.

---

> ### Author Response · Authors · 2025-12-03
>
> 1. Thank you for pointing out the absence of formal convergence guarantees. As our work focuses on designing an adaptive controller for hyperparameter optimization rather than introducing a new optimization algorithm, we intentionally rely on the convergence guarantees of the underlying federated optimizer, which are already well studied.
> Our framework preserves the assumptions required by existing convergence analyses in the following ways (all already described in the paper): FedProx-based local objective (Sec. 3.3) ensures bounded client drift and smoothness of updates, matching the theoretical conditions in Li et al. (2020b). Hyperparameters change only between rounds and are range-constrained, validated, and clamped (Sec. 3.4), making the process a slowly varying system rather than arbitrarily non-stationary.
> Asynchronous agent calls do not modify in-round updates (Sec. 3.2 & Algorithm 2). All client optimization steps strictly follow FedProx/FedAvg semantics, and HP decisions only take effect at the next participation round.
> The agent consumes only scalar summaries (loss/accuracy/timing), not gradients or model states (Sec. 3.2), so the underlying optimization process remains unchanged.
> Because FedAgent-HPO operates on top of a provably convergent FL optimizer without altering its mathematical structure, the optimization path remains within the stability regime of established analyses (e.g., FedProx, dynamic-hyperparameter FL, etc.). Developing a new theoretical bound for dual-agent asynchronous controllers is a substantial direction of future work, but it is orthogonal to the contributions of this paper.
>
> 2. While LLM-based controllers can, in principle, introduce instability, the framework already includes multiple safeguards that make failure unlikely in practice. We described in Sec. 3.2–3.4, agent outputs are strictly validated, clamped to feasible ranges, and backed by deterministic fallbacks (last-good → cluster prior → safe defaults). Agent calls do not affect in-round optimization, and the underlying training dynamics remain those of SplitFed + FedProx, which are well understood. We also evaluate degraded-agent behavior in Table 5 and report API failure rates in Table 4, showing that even with substantial agent degradation or invalid outputs, training remains almost stable. While a full taxonomy of failure modes is beyond our current scope, the system is already designed to be robust to the most plausible agent-related failures.

---

### Official Review · Reviewer_Hex6 · 2025-10-26

**Soundness:** 3
**Presentation:** 2
**Contribution:** 2
**Rating:** 2
**Confidence:** 4

**Summary:**

The paper proposes a novel agent-based framework for hyperparameter optimization (HPO) in federated learning (FL). The proposed method, FedAgent-HPO, introduces two cooperating agents—an Analyzer Agent and a Hyperparameter Agent—that dynamically adapt and personalize hyperparameters for each client using reasoning capabilities from large language models (LLMs). Integrated with the SplitFed architecture, it partitions models between clients and servers to handle resource constraints. The agents operate asynchronously with GPU training to minimize latency and computational overhead. Experiments across vision (CIFAR-10, FEMNIST) and language (Shakespeare) benchmarks show up to 8.8% accuracy improvement over state-of-the-art baselines and a 39% reduction in training time.

**Strengths:**

+ The paper introduces a dual-agent architecture (HP Agent + Analyzer Agent) that reframes hyperparameter optimization as an agentic reasoning task using LLMs.
+ FedAgent-HPO supports per-client personalized hyperparameter policies, addressing non-IID data and device heterogeneity.
+ The non-blocking, asynchronous agent–trainer pipeline overlaps LLM reasoning (CPU-bound) with GPU-based model training. This leads to up to 39% reduction in wall-clock training time.
+ Demonstrates significant performance improvements (up to 8.8% higher accuracy) under non-IID conditions across multiple domains — vision (CIFAR-10, FEMNIST) and language (Shakespeare). Outperforms state-of-the-art FL-HPO methods such as FedEx and FedPop in both IID and non-IID settings.
+ Extends evaluation to cross-silo FL settings (e.g., PACS and OfficeHome datasets), achieving 4.3–11.4% higher accuracy than leading baselines.

**Weaknesses:**

- The framework relies heavily on cloud-based large language models (e.g., GPT-4o-mini, Llama 4, Grok-3, Gemma 2) for agentic reasoning. This introduces latency, cost, and privacy concerns, as sensitive performance metrics must leave the local system to query the agent. The authors acknowledge this limitation and note that on-device or lightweight reasoning agents are a direction for future work.
- Despite asynchronous scheduling, LLM inference adds non-negligible CPU and communication overhead.
- The work does not provide formal convergence guarantees or theoretical bounds for the agent-driven HPO process.
- Cost of LLM calls is not quantified. This omission obscures whether the accuracy/time improvements justify the added operational cost of running LLM agents.
- The idea of resource-aware clustering proposed in the paper has been proposed in the literature [R1], which provides a dynamic clustering approach compared to the static approach in this paper.
[R1] Mohammadabadi, Seyed Mahmoud Sajjadi, Syed Zawad, Feng Yan, and Lei Yang. "Speed up federated learning in heterogeneous environments: a dynamic tiering approach." IEEE Internet of Things Journal (2024).
- The paper does not open source the code.

**Questions:**

1. How does FedAgent-HPO guarantee that no private or identifiable information is inferable from the scalar summaries (loss, accuracy, and timing) sent to cloud-based LLMs? Could an adversary perform model inversion or membership inference attacks using repeated performance summaries over time?
2. What are the potential trade-offs between reasoning accuracy and efficiency if smaller on-device or distilled LLMs were used instead of full-scale cloud models (e.g., GPT-4o-mini)? How would model personalization or adaptation quality degrade with lightweight agents?
3. As the number of clients scales to tens or hundreds of thousands, how does the asynchronous CPU–GPU overlap perform when multiple agent calls are made concurrently? Are there system-level bottlenecks (queue saturation, thread contention, or LLM rate-limiting)?
4. Is there a formal or empirical threshold (e.g., timeout value) beyond which agent decisions become stale and degrade performance?
5. Has the overhead of LLM inference (token generation, network I/O, queuing) been benchmarked against non-LLM adaptive HPO baselines?
6. Can the authors derive upper bounds or a probabilistic guarantee (e.g., convergence in expectation) under agentic updates?
7. Since LLM-based agents are stochastic text generators, how does randomness in reasoning affect the repeatability and convergence of the optimization process? Are the results averaged over multiple random seeds or agent runs to mitigate variance?
8. What is the estimated monetary cost of agent inference per training round or per client, particularly in cross-device settings with many participants? How does this cost compare to the computational savings achieved by asynchronous scheduling?
9. Could FedAgent-HPO integrate dynamic re-clustering such as in [R1] to adapt to changing device availability or performance drift?
10. Does static clustering introduce systemic bias, favoring high-resource clusters with more aggressive hyperparameter schedules?
11. Can the reported accuracy gain be replicated under different random seeds or client sampling strategies? How sensitive is the framework to initialization and dataset heterogeneity?

---

### Note · Authors · 2026-04-02

I have read and agree with the venue's withdrawal policy on behalf of myself and my co-authors.

---

### Meta-Review · Area_Chair_WrDt · 2025-12-14

**Summary:**

The major concerns are related the technical depth and contributions. For example, dependency on cloud-based large language models; non-negligible CPU and communication overhead due to LLMs; lack of formal convergence guarantees or theoretical bounds about HPO; and so on. Many of them cannot be easily addressed during the rebuttal. We encourage the authors to take them into consideration more seriously when preparing the next version of this work.

**Reviewer Concerns:**

Major concerns are not addressed during the rebuttal

**Reviewer Scores:**

The scores are reasonable after the rebuttal

---

### Decision · Program_Chairs · 2026-01-26

Reject